# Inverse-Free Sparse Variational Gaussian Processes

**Stefano Cortinovis**\*
University of Oxford

**Laurence Aitchison**
University of Bristol

**James Hensman**
Microsoft Research

**Stefanos Eleftheriadis**
No affiliation

**Mark van der Wilk**
University of Oxford

## 1  Introduction

Gaussian processes (GPs) [24] are priors over functions with many desirable properties, but also with many computational difficulties. Despite progress, the popular variational method [20, 8] is still complicated by the need to invert/decompose the kernel matrix, which is **1)** difficult in low-precision, **2)** less parallelisable than matmuls, and **3)** incurs cubic time cost in model capacity.

Van der Wilk et al. [23, 22] proposed variational bounds that involve only matmuls by introducing an additional variational parameter $\mathbf{T} \in \mathbb{R}^{M \times M}$, which becomes the required inverse at the optimum. This turns the inversion operation into an optimisation problem. Although, in theory, this would not introduce performance gaps, *practical* success requires the optimisation of $\mathbf{T}$ to be fast enough to keep up with the changing kernel throughout training. Unfortunately, typical deep learning optimisers were shown to converge unusably slowly.

We solve this through **1)** a preconditioner for a variational parameter, **2)** a tailored update for $\mathbf{T}$ based on natural gradients or Newton's method, and **3)** a stopping criterion to determine the number of updates. This leads to an inverse-free method on-par with existing approaches on an iteration basis, with low-precision computation and wall-clock speedups being the next step.

## 2  Sparse variational Gaussian process (SVGP) parameterisations

We consider learning a function $f : \mathbb{R}^D \to \mathbb{R}$ through Bayesian inference, with a prior $f(\cdot) \sim \mathcal{GP}(0, k_\theta(\cdot, \cdot))$ and an arbitrary factorised likelihood with density $p(\mathbf{y}|\mathbf{f}) = \prod_{n=1}^N p(y_n|f(\mathbf{x}_n), \theta)$. To make predictions, we need to approximate the maximum marginal likelihood hyperparameters $\theta^* = \operatorname{argmax}_\theta \log p(\mathbf{y}|\theta)$, and the posterior $p(f|\mathbf{y})$. To do so, the predominant approach is to use sparse variational Gaussian processes (SVGPs) [8], which select an approximate posterior distribution $q(f)$ by minimising the KL-divergence to the true posterior $p(f|\mathbf{y})$. The approximate posterior is constructed by conditioning the prior on $M \ll N$ inducing points [15], at input locations $\mathbf{Z} \in \mathbb{R}^{M \times D}$ and output values $\mathbf{u} = f(\mathbf{Z})$, resulting in a variational posterior $q(f) = \int p(f_{\neq \mathbf{u}}|\mathbf{u})q(\mathbf{u})\mathrm{d}\mathbf{u}$ (see [21, 12]). Following variational inference [6], we minimise $\mathrm{KL}[q(f)||p(f|\mathbf{y})]$ by maximising the evidence lower bound (ELBO) with respect to $\mathbf{Z}$ and parameters of $q(\mathbf{u})$. In this case, the ELBO becomes [8]

$$\mathcal{L} = \sum_{n=1}^N \mathbb{E}_{\mathcal{N}(\mu_n, \sigma_n^2)} \left[ \log p(y_n|f(\mathbf{x}_n)) \right] - \mathrm{KL} \left[ q(\mathbf{u})||p(\mathbf{u}) \right], \tag{1}$$

where $\mu_n$ and $\sigma_n$ are, respectively, the predictive mean and variance at the input location $\mathbf{x}_n$. The choice of the parameterisation of the Gaussian distribution $q(\mathbf{u})$ can have a significant impact on the optimisation stability and the tightness of the ELBO. Below, we briefly review two common

---

\*Correspondence to: `cortinovis@stats.ox.ac.uk`

Workshop on Bayesian Decision-making and Uncertainty, 38th Conference on Neural Information Processing Systems (NeurIPS 2024).

parameterisations, as well as their relationship with the inverse-free variational bound of van der Wilk et al. [22]. We highlight with red colour the terms in $\mu_n$, $\sigma_n^2$ and $\mathrm{KL}[q(\mathbf{u})||p(\mathbf{u})]$ that involve matrix decompositions.

**Marginal parameterisation (M-SVGP).** The most common freeform $q(\mathbf{u}) = \mathcal{N}(\mathbf{m}, \mathbf{S})$ results in

$$\mu_n = \mathbf{k}_{n\mathbf{u}}\mathbf{K_{uu}^{-1}}\mathbf{m}, \qquad \sigma_n^2 = k_{nn} - \mathbf{k}_{n\mathbf{u}}\mathbf{K_{uu}^{-1}}\mathbf{k}_{\mathbf{u}n} + \mathbf{k}_{n\mathbf{u}}\mathbf{K_{uu}^{-1}}\mathbf{S}\mathbf{K_{uu}^{-1}}\mathbf{k}_{\mathbf{u}n}, \tag{2}$$

$$\mathrm{KL}\left[q(\mathbf{u})||p(\mathbf{u})\right] = \frac{1}{2}\left(\mathrm{tr}(\mathbf{K_{uu}^{-1}}\mathbf{S}) + \mathbf{m}^\top\mathbf{K_{uu}^{-1}}\mathbf{m} - M + \log|\mathbf{K_{uu}}| - \log|\mathbf{S}|\right), \tag{3}$$

where $\mathbf{K_{uu}} = k(\mathbf{Z}, \mathbf{Z})$, $\mathbf{k}_{\mathbf{u}n} = \mathbf{k}_{n\mathbf{u}}^\top = k(\mathbf{Z}, \mathbf{x}_n)$, and $k_{nn} = k(\mathbf{x}_n, \mathbf{x}_n)$.

**Likelihood parameterisation (L-SVGP).** Panos et al. [14] suggest to reparameterise $q(\mathbf{u}) = \mathcal{N}(\mathbf{m}, \mathbf{S})$ with $\mathbf{m} = \mathbf{K_{uu}}\tilde{\mathbf{m}}$ and $\mathbf{S} = (\mathbf{K_{uu}^{-1}} + \tilde{\mathbf{S}}^{-1})^{-1}$, where $\tilde{\mathbf{S}}$ is PD diagonal. This results in

$$\mu_n = \mathbf{k}_{n\mathbf{u}}\tilde{\mathbf{m}}, \quad \sigma_n^2 = k_{nn} - \mathbf{k}_{n\mathbf{u}}\tilde{\mathbf{K}}^{-1}\mathbf{k}_{\mathbf{u}n} =: \sigma_n^{2(\mathrm{L})}, \tag{4}$$

$$\mathrm{KL}\left[q(\mathbf{u})||p(\mathbf{u})\right] = \frac{1}{2}\left(-\mathrm{tr}(\tilde{\mathbf{K}}^{-1}\mathbf{K_{uu}}) + \tilde{\mathbf{m}}^\top\mathbf{K_{uu}}\tilde{\mathbf{m}} - M + \log|\tilde{\mathbf{K}}| - \log|\tilde{\mathbf{S}}|\right), \tag{5}$$

where $\tilde{\mathbf{K}} = \mathbf{K_{uu}} + \tilde{\mathbf{S}}$. Inverting $\tilde{\mathbf{K}}$ instead of $\mathbf{K}$ is more stable and does not need *jitter* terms, due to having lower-bounded minimum eigenvalue thanks to $\tilde{\mathbf{S}}$.

**Inverse-free likelihood parameterisation (R-SVGP).** Van der Wilk et al. [22] develop an inverse-free bound for L-SVGP, starting by upper bounding the predictive variance as

$$\sigma_n^{2(\mathrm{L})} \leq k_{nn} + \mathbf{k}_{n\mathbf{u}}(\mathbf{T}\tilde{\mathbf{K}}\mathbf{T} - 2\mathbf{T})\mathbf{k}_{\mathbf{u}n} =: U_n, \tag{6}$$

where $\mathbf{T} \in \mathbb{R}^{M \times M}$, and with equality when $\mathbf{T} = \tilde{\mathbf{K}}^{-1}$. Working backwards, they then find that, if the variance $\mathbf{S}$ of the marginal parameterisation is reparameterised as $\mathbf{S} = \mathbf{K_{uu}} - \mathbf{K_{uu}}(2\mathbf{T} - \mathbf{T}\tilde{\mathbf{K}}\mathbf{T})\mathbf{K_{uu}}$, the predictive variance matches the upper bound $\sigma_n^2 = U_n$. This improves over the previous inverse-free bound for M-SVGP [23] by allowing for a closed-form inverse-free bound for the KL term

$$\mathrm{KL}\left[q(\mathbf{u})||p(\mathbf{u})\right] \leq \frac{1}{2}\left(\mathrm{tr}((\mathbf{T}\tilde{\mathbf{K}}\mathbf{T} - 2\mathbf{T})\mathbf{K_{uu}}) + \tilde{\mathbf{m}}^\top\mathbf{K_{uu}}\tilde{\mathbf{m}}\right.$$
$$\left. +\mathrm{tr}(\tilde{\mathbf{K}}\mathbf{T}) - M - \log|\mathbf{T}| - \log|\tilde{\mathbf{S}}|\right), \tag{7}$$

with equality when $\mathbf{T} = \tilde{\mathbf{K}}^{-1}$. Putting all this together leads to a new ELBO that **1)** depends on the additional parameter $\mathbf{T}$, **2)** is free of matrix decompositions, **3)** is always a lower bound to the L-SVGP ELBO and **4)** recovers the L-SVGP solution when $\mathbf{T} = \tilde{\mathbf{K}}^{-1}$.

**Alternative inverse-free parameterisations.** While we do not pursue the idea further in this work, we note that the R-SVGP construction, whereby one exploits an inverse-free upper bound to the predictive variance, is more general and may be applied to parameterisations other than L-SVGP. For instance, starting from M-SVGP, reparameterise $\mathbf{m} = \mathbf{K_{uu}}\mathbf{R}\tilde{\mathbf{m}}$ and $\mathbf{S} = \mathbf{K_{uu}}\mathbf{R}\tilde{\mathbf{S}}\mathbf{R}^\top\mathbf{K_{uu}}$, where $\tilde{\mathbf{S}}$ is PD and $\mathbf{R}$ is lower triangular. This results in

$$\mu_n = \mathbf{k}_{n\mathbf{u}}\mathbf{R}\tilde{\mathbf{m}}, \quad \sigma_n^2 = k_{nn} - \mathbf{k}_{n\mathbf{u}}(\mathbf{K_{uu}^{-1}} - \mathbf{R}\tilde{\mathbf{S}}\mathbf{R}^\top)\mathbf{k}_{\mathbf{u}n}. \tag{8}$$

Similarly to Eq. (6), the predictive variance can be upper bounded as

$$\sigma_n^2 \leq k_{nn} + \mathbf{k}_{n\mathbf{u}}\mathbf{R}(\mathbf{R}^\top\mathbf{K_{uu}}\mathbf{R} - 2\mathbf{I} + \tilde{\mathbf{S}})\mathbf{R}^\top\mathbf{k}_{\mathbf{u}n}, \tag{9}$$

with equality when $\mathbf{R} = \mathrm{chol}(\mathbf{K_{uu}^{-1}})$. Working backwards, the upper bound above is equal to the predictive variance induced by the reparameterisation $\mathbf{S} = \mathbf{K_{uu}} + \mathbf{K_{uu}}\mathbf{R}(\mathbf{R}^\top\mathbf{K_{uu}}\mathbf{R} - 2\mathbf{I} + \tilde{\mathbf{S}})\mathbf{R}^\top\mathbf{K_{uu}}$. The KL term implied by this choice of $\mathbf{m}$ and $\mathbf{S}$ admits the upper bound

$$\mathrm{KL}[q(\mathbf{u})||p(\mathbf{u})] \leq \frac{1}{2}\left(\mathrm{tr}\left((\tilde{\mathbf{S}} - 2\mathbf{I})\mathbf{R}^\top\mathbf{K_{uu}}\mathbf{R}\right) + \tilde{\mathbf{m}}^\top\mathbf{R}^\top\mathbf{K_{uu}}\mathbf{R}\tilde{\mathbf{m}}\right.$$
$$\left. - \log|\tilde{\mathbf{S}}| + \mathrm{KL}\left[\mathcal{N}(\mathbf{0}, \mathbf{R}\mathbf{R}^\top)||\mathcal{N}(\mathbf{0}, \mathbf{K_{uu}^{-1}})\right]\right), \tag{10}$$

with equality when $\mathbf{R} = \mathrm{chol}(\mathbf{K_{uu}^{-1}})$. The resulting ELBO, which depends on the additional parameter $\mathbf{R}$, recovers the popular whitened SVGP parameterisation [9] at its optimum $\mathbf{R} = \mathrm{chol}(\mathbf{K_{uu}^{-1}})$. Morever, while Eq. (10) still contains an inverse in the inner KL term, we show in the next section how one can efficiently minimise the latter with respect to $\mathbf{R}$, thereby making the required update for the rest of the parameters effectively inverse-free.

# 3 Optimising the inverse-free variational bound

The effectiveness of the R-SVGP parameterisation relies on the ability to optimise $\mathbf{T}$ close to its optimal value. In their experiments, Van der Wilk et al. [22] parameterise $\mathbf{T}$ through its Cholesky factor $\mathbf{L}$ as $\mathbf{T} = \mathbf{LL}^\top$, which they update jointly with the other model parameters using Adam [10]. However, they report that this results in poor optimisation performance even with simple datasets. Here, we solve this problem with a tailored optimisation procedure.

## 3.1 Inverting matrices iteratively with natural gradients

Given a symmetric PD matrix $\mathbf{A}$, the Cholesky factor $\mathbf{L}_{\mathbf{A}^{-1}}$ of its inverse $\mathbf{A}^{-1}$ can be written as

$$\mathbf{L}_{\mathbf{A}^{-1}} = \arg\min_{\mathbf{L}} \ell_{\mathbf{A}}(\mathbf{L}), \qquad \ell_{\mathbf{A}}(\mathbf{L}) := \mathrm{KL}\left[\mathcal{N}(\mathbf{0}, \mathbf{LL}^\top)||\mathcal{N}(\mathbf{0}, \mathbf{A}^{-1})\right]. \tag{11}$$

Natural gradient (NG) updates are known to converge quickly on KL objectives [3]. In particular, we descend along the natural gradient $\tilde{\nabla}\ell_{\mathbf{A}} = \mathbf{F}^{-1}(\nabla \ell_{\mathbf{A}})$, which is found by preconditioning the standard gradient by the inverse Fisher information matrix $\mathbf{F} = -\mathbb{E}_{\mathcal{N}(x;\mathbf{0},\mathbf{LL}^\top)}\left[\nabla_{\mathbf{L}}^2 \log \mathcal{N}(\mathbf{x}; 0, \mathbf{LL}^\top)\right]$. For Eq. (11), the natural gradient admits a simple closed-form expression, derived in Appendix A.

**Proposition 1.** *Let $\ell(\mathbf{L})$ be as in Eq. (11). Then,*

$$\tilde{\nabla}\ell_{\mathbf{A}} = \mathbf{L}\left[\mathrm{tril}(\mathbf{L}^\top \mathbf{A} \mathbf{L}) - \frac{1}{2}(\mathbf{I} + \mathrm{diag}(\mathbf{L}^\top \mathbf{A} \mathbf{L}))\right], \tag{12}$$

*where $\mathrm{tril}(\cdot)$ and $\mathrm{diag}(\cdot)$ return the lower triangular part and the diagonal of a matrix.*

With this, we can optimise the R-SVGP bound by alternating between two steps:

1) **NG step**: for the current $\tilde{\mathbf{K}}$, update $\mathbf{L}$ by applying $\mathbf{L}_{t+1} = \mathbf{L}_t - \gamma \tilde{\nabla}_{\mathbf{L}} \ell_{\tilde{\mathbf{K}}}$, $t^*$ times.

2) **Adam step**: given the current value of $\mathbf{L}$, update the other model parameters by applying an Adam update to maximise the R-SVGP bound.

Crucially, the expression for the natural gradient in Eq. (12) does not involve matrix decompositions, thereby maintaining the overall optimisation procedure *inverse-free*. Interestingly, such an update for $\mathbf{L}$ is related to the well-known Newtonian iteration for computing the inverse of a matrix [5].

The R-SVGP parameterisation introduces some slack to the L-SVGP objective through the upper bound on the predictive variance $\sigma_n^{2(\mathrm{L})}$ in Eq. (6). For Gaussian likelihoods, we can monitor such a bias to choose the number $t^*$ of NG updates for $\mathbf{T}$. This approach is similar in spirit to several stopping criteria used for conjugate gradient approximations of Gaussian processes [11, 4]. We start by deriving the following lower bound on $\sigma_n^{2(\mathrm{L})}$, which is proved in Appendix B.

**Proposition 2.** *Let $\sigma_n^{2(L)}$ be as in Eq. (4). Since $\tilde{\mathbf{S}}$ is diagonal and PD, without loss of generality, we can rewrite it as $\tilde{\mathbf{S}} = \tilde{\mathbf{S}}' + \sigma^2 \mathbf{I}$, where $\tilde{\mathbf{S}}'$ is diagonal and PD. Then,*

$$\sigma_n^{2(L)} \geq k_{nn} - \frac{1}{\sigma^2}\left(\mathbf{k}_{n\mathbf{u}}\mathbf{k}_{\mathbf{u}n} - 2\mathbf{k}_{n\mathbf{u}}\mathbf{T}\tilde{\mathbf{K}}_-\mathbf{k}_{\mathbf{u}n} + \mathbf{k}_{n\mathbf{u}}\mathbf{T}\tilde{\mathbf{K}}_-\tilde{\mathbf{K}}\mathbf{T}\mathbf{k}_{\mathbf{u}n}\right) =: L_n, \tag{13}$$

*where $\tilde{\mathbf{K}}_- = \mathbf{K}_{\mathbf{uu}} + \tilde{\mathbf{S}}'$, so that $\tilde{\mathbf{K}} = \tilde{\mathbf{K}}_- + \sigma^2 \mathbf{I}$.*

By subtracting the upper and lower bounds on $\sigma_n^{2(\mathrm{L})}$, we find the quantity

$$G_n := U_n - L_n = \frac{1}{\sigma^2}||(\mathbf{I} - \tilde{\mathbf{K}}\mathbf{T})\mathbf{k}_{\mathbf{u}n}||^2. \tag{14}$$

Moreover, for Gaussian likelihoods, the predictive variance $\sigma_n^2$ enters the expectation component of the ELBO through the term $-\frac{1}{2\sigma_{\mathrm{obs}}^2}\sum_{n=1}^N \sigma_n^2$, where $\sigma_{\mathrm{obs}}^2$ is the likelihood variance. Therefore, the stopping criterion $G := \sum_{n=1}^N G_n \leq 2\sigma_{\mathrm{obs}}^2 \epsilon$ ensures that the gap between the R-SVGP and L-SVGP bounds due to the approximation of $\sigma_n^{2(\mathrm{L})}$ is at most $\epsilon$.

## 3.2 Improving the performance of L-SVGP with inducing mean preconditioning

When $\mathbf{T}$ is optimised perfectly, the R-SVGP bound coincides with the L-SVGP bound. Hence, the ability for L-SVGP to reach competitive performance is a necessary condition for R-SVGP to do the same. Unfortunately, we find across a range of experiments that the L-SVGP bound is more difficult to optimise than the standard M-SVGP bound, often leading to worse predictive performance. We identify the cause of this issue in the inducing mean parameterisation $\mathbf{m} = \mathbf{K_{uu}}\tilde{\mathbf{m}}$ and we propose to address it by pre-multiplying $\tilde{\mathbf{m}}$ with the matrix $\tilde{\mathbf{K}}^{-1} = (\mathbf{K_{uu}} + \tilde{\mathbf{S}})^{-1}$. The intuition behind this change is that, in the case where the data points $\mathbf{X}$ and the inducing inputs $\mathbf{Z}$ coincide, the resulting posterior mean $\mu_n = \mathbf{k}_{n\mathbf{u}}(\mathbf{K_{uu}} + \tilde{\mathbf{S}})^{-1}\tilde{\mathbf{m}}$ is precisely in the form of the true posterior mean of a GP with Gaussian likelihood, for which $\tilde{\mathbf{S}} = \sigma_{\text{obs}}^2\mathbf{I}$ and $\tilde{\mathbf{m}} = \mathbf{y}$ [14, 2]. However, in the R-SVGP bound, we cannot simply pre-multiply $\tilde{\mathbf{m}}$ with $\tilde{\mathbf{K}}^{-1}$ without defeating the purpose of it being inverse-free. Instead, we use $\mathbf{T}$ as an approximate preconditioner. In doing so, we also hardcode the custom gradient of $\mathbf{T}\tilde{\mathbf{m}}$ wrt $\tilde{\mathbf{K}}$ as if $\mathbf{T}$ was exactly equal to $\tilde{\mathbf{K}}^{-1}$. Our hope is that, when $\mathbf{T}$ is sufficiently well optimised, this will lead to similar benefits.

## 4 Results

To assess the effectiveness of the NG updates for $\mathbf{T}$ (Section 3.1) and the inducing mean preconditioning (Section 3.2), we compare versions of the R-SVGP bound that take advantage of either or both of these tools. The main baseline is the R-SVGP bound without preconditioning and where $\mathbf{T}$ is trained with Adam, as in van der Wilk et al. [22]. In this context, the optimisation of the R-SVGP bound is successful when the resulting model matches the performance of the one trained with the corresponding L-SVGP bound. In all comparisons, we include the M-SVGP parameterisation, which is the de facto standard baseline for stochastic variational GP approximations. The setup of all the experiments discussed below is presented in Appendix C.

The benefits of the tools proposed in Section 3 are apparent even in toy datasets. Fig. 1 shows training loss traces for each bound optimised on the SNELSON regression and BANANA classification datasets. In both cases, the R-SVGP bounds with NG updates (i.e. R-SVGP-N and R-SVGP-NP) match

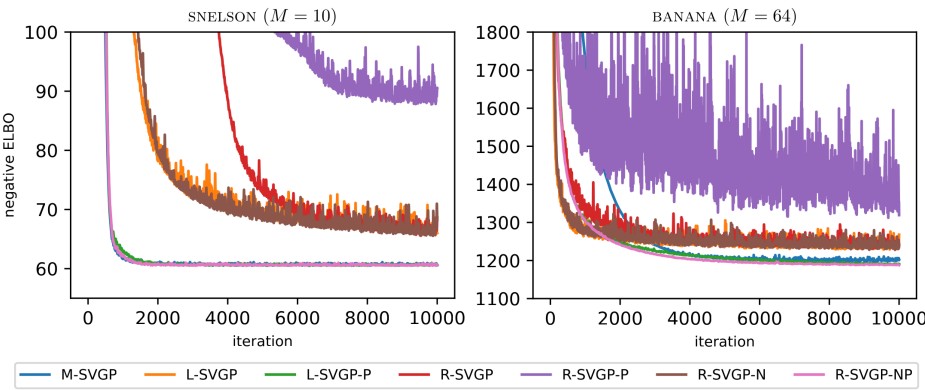

Figure 1: Loss traces on SNELSON and BANANA datasets. The P and N suffixes in the legend names refer to the use of inducing mean preconditioning (Section 3.2) or NG updates (Section 3.1), respectively.

the performance of the corresponding L-SVGP bounds (i.e. L-SVGP and L-SVGP-P, respectively). Conversely, the bounds trained only with Adam suffer from either much slower convergence (R-SVGP) or worse results (R-SVGP-P) than their NG-counterparts. Furthermore, preconditioning is crucial for the L-SVGP and R-SVGP bounds to match the performance of M-SVGP. However, it is only beneficial for R-SVGP when coupled with the NG updates. This is explained by the fact that, as discussed in Section 3.2, preconditioning is expected to help only when $\mathbf{T}$ is a good enough approximation to $\tilde{\mathbf{K}}$. As shown in Appendix D.1, this is usually only the case when we use NG updates.

Larger datasets make the performance gap between RSVGP-NP and the other versions of the RSVGP bound even more significant. To show this, we evaluate our method on four UCI regression datasets. As shown in Appendix D.2 and Appendix D.3, the R-SVGP-NP bound recovers the performance of L-SVGP-P on all datasets, regardless of whether the inducing locations $\mathbf{Z}$ are fixed and trained. When $\mathbf{Z}$ is fixed, we find that a single NG update with step size one at every iteration is enough. On the other hand, when $\mathbf{Z}$ is trained, multiple NG updates are required, which represents a use case for the stopping criterion derived in Section 3.1. In particular, at each iteration, we employ an exponentially increasing schedule for the NG step size and stop optimising $\mathbf{T}$ when $G$ falls below a predetermined threshold. In all cases, RSVGP-NP is the only version of the RSVGP bound that is able to match the performance of M-SVGP and LSVGP-P, demonstrating the effectiveness of our approach for optimising the inverse-free bound.

## 5   Discussion

We introduced new and effective techniques to optimise the inverse-free bound of van der Wilk et al. [22]. For the first time, our method allows to successfully train a sparse variational GP model on realistic datasets without the computation of matrix decompositions, and we are extremely excited about the next steps in this direction. In particular, while we find that multiple NG updates for $\mathbf{T}$ are required when training $\mathbf{Z}$, we only experiment with a basic exponential schedule for the NG step size. We expect that a more tailored approach, possibly using a log-linear schedule [18], may further improve stability and robustness. Furthermore, since our method only employs matmuls, it should be possible to implement it using only single-precision arithmetic. A careful runtime benchmark of our method, both using single and double precision floats, is required to assess the computational benefit of using the inverse-free bound to train variational GPs. Lastly, our method could be extended to derive and optimise inverse-free versions of other SVGP parameterisations, as well as other GP architectures based on the SVGP framework, such as deep GPs [17]. All in all, we believe that our results, while preliminary, hint at the intriguing possibility of building complex GP models that are more scalable and better suited to modern hardware.

## Acknowledgements

The authors would like to acknowledge the use of the University of Oxford Advanced Research Computing (ARC) facility [16] in carrying out this work. Stefano Cortinovis is supported by the EPSRC Centre for Doctoral Training in Modern Statistics and Statistical Machine Learning (EP/S023151/1).

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

## A   Proof of Proposition 1

*Proof.* Given the decomposition for our variational parameter $\mathbf{T} = \mathbf{L}\mathbf{L}^\top$, our goal is to find the optimal $\mathbf{L}$ so that $\mathbf{T} = (\mathbf{K_{uu}} + \tilde{\mathbf{S}})^{-1}$, without explicitly computing the inverse. We consider the problem of finding the Cholesky factor of the inverse of a matrix $\mathbf{A}$ via:

$$\mathbf{L_{A^{-1}}} = \arg\min_{\mathbf{L}} \ell_{\mathbf{A}}(\mathbf{L}) \qquad \ell_{\mathbf{A}}(\mathbf{L}) := \mathrm{KL}\left[\mathcal{N}(\mathbf{0}, \mathbf{L}\mathbf{L}^\top)\|\mathcal{N}(\mathbf{0}, \mathbf{A}^{-1})\right]. \tag{15}$$

The above optimisation problem describes the update we take in the direction of the natural gradient:

$$\tilde{\mathbf{g}} = \mathbf{F}^{-1}\mathbf{g}, \tag{16}$$

where $\mathbf{g}$ and $\tilde{\mathbf{g}}$ are column vectors denoting the gradient and the natural gradient of $\ell_{\mathbf{A}}$ wrt $\mathbf{L}$, while we denote with $\mathbf{F}$ the Fisher information matrix of the distribution $p(\mathbf{x}; \mathbf{L}) = \mathcal{N}(\mathbf{x}|\mathbf{0}, \mathbf{L}\mathbf{L}^\top)$ parameterised by $\mathbf{L}$:

$$\mathbf{F} = -\mathbb{E}_p\left[\frac{\partial^2}{\partial \mathrm{vec}(\mathbf{L})\, \partial \mathrm{vec}(\mathbf{L}^\top)} \log p(\mathbf{x}; \mathbf{L})\right], \tag{17}$$

with $\mathrm{vec}(\cdot)$ the column-wise vectorisation operator.

We start by computing the gradient of the loss $\nabla \ell_{\mathbf{A}}$ wrt $\mathbf{L}$. As in Eq. (11),

$$\ell_{\mathbf{A}}(\mathbf{L}) = \mathrm{KL}\left[p(\mathbf{x}; \mathbf{L})\|\mathcal{N}(\mathbf{0}, \mathbf{A}^{-1})\right] = \frac{1}{2}\left(\mathrm{tr}(\mathbf{A}\mathbf{L}\mathbf{L}^\top) - \log|\mathbf{L}\mathbf{L}^\top|\right) + c_1, \tag{18}$$

where $c_1$ is a constant that does not depend on $\mathbf{L}$. The first differential of $\ell_{\mathbf{A}}$ with respect to $\mathbf{L}$ is given by

$$\mathrm{d}\,\ell_{\mathbf{A}} = \frac{1}{2}\underbrace{\mathrm{d}\,\mathrm{tr}(\mathbf{A}\mathbf{L}\mathbf{L}^\top)}_{(a)} - \frac{1}{2}\underbrace{\mathrm{d}\log|\mathbf{L}\mathbf{L}^\top|}_{(b)}. \tag{19}$$

Term $(a)$ can be computed as

$$\begin{aligned}
\mathrm{d}\,\mathrm{tr}(\mathbf{A}\mathbf{L}\mathbf{L}^\top) &= \mathrm{tr}(\mathbf{A}\,\mathrm{d}(\mathbf{L}\mathbf{L}^\top)) \\
&= \mathrm{tr}(\mathbf{A}\,\mathrm{d}\mathbf{L}\mathbf{L}^\top + \mathbf{A}\mathbf{L}\,\mathrm{d}\mathbf{L}^\top) \\
&= \mathrm{tr}(\mathbf{L}^\top\mathbf{A}\,\mathrm{d}\mathbf{L}) + \mathrm{tr}(\mathbf{A}\mathbf{L}\,\mathrm{d}\mathbf{L}^\top) \\
&= 2\,\mathrm{vec}(\mathbf{A}\mathbf{L})^\top\,\mathrm{vec}(\mathrm{d}\mathbf{L}),
\end{aligned} \tag{20}$$

where the last equality holds because $\mathbf{A}$ is symmetric. Term $(b)$ is given by

$$\begin{aligned}
\mathrm{d}\log|\mathbf{L}\mathbf{L}^\top| &= \frac{1}{|\mathbf{L}\mathbf{L}^\top|}\mathrm{d}\,|\mathbf{L}\mathbf{L}^\top| \\
&= \frac{1}{|\mathbf{L}\mathbf{L}^\top|}|\mathbf{L}\mathbf{L}^\top|\,\mathrm{tr}(\mathbf{L}^{-\top}\mathbf{L}^{-1}\,\mathrm{d}(\mathbf{L}\mathbf{L}^\top)) \\
&= \mathrm{tr}(\mathbf{L}^{-\top}\mathbf{L}^{-1}\,\mathrm{d}\mathbf{L}\mathbf{L}^\top + \mathbf{L}^{-\top}\mathbf{L}^{-1}\mathbf{L}\,\mathrm{d}\mathbf{L}^\top) \\
&= \mathrm{tr}(\mathbf{L}^{-1}\,\mathrm{d}\mathbf{L}) + \mathrm{tr}(\mathbf{L}^{-\top}\,\mathrm{d}\mathbf{L}^\top) \\
&= 2\,\mathrm{vec}(\mathbf{L}^{-\top})^\top\,\mathrm{vec}(\mathrm{d}\mathbf{L}).
\end{aligned} \tag{21}$$

Putting it all together, the differential of $\ell_{\mathbf{A}}$ takes the form

$$\mathrm{d}\,\ell_{\mathbf{A}} = (\mathrm{vec}(\mathbf{A}\mathbf{L}) + \mathrm{vec}(\mathbf{L}^{-\top}))^\top\,\mathrm{vec}(\mathrm{d}\mathbf{L}), \tag{22}$$

which implies that

$$\mathbf{g} = \mathrm{vec}(\mathbf{A}\mathbf{L}) - \mathrm{vec}(\mathbf{L}^{-\top}). \tag{23}$$

We now move to computing the Fisher information matrix $\mathbf{F}$ of $p(\mathbf{x}; \mathbf{L})$ wrt $\mathbf{L}$. We have that

$$\log p(\mathbf{x}; \mathbf{L}) = -\frac{1}{2}\log|\mathbf{L}\mathbf{L}^\top| - \frac{1}{2}\mathbf{x}^\top\mathbf{L}^{-\top}\mathbf{L}^{-1}\mathbf{x} + c_2, \tag{24}$$

where $c_2$ is a constant that does not depend on $\mathbf{L}$. The quantity of interest is the expected value of the second differential of $\log p(\mathbf{x}; \mathbf{L})$ wrt $p$, which takes the form

$$\mathbb{E}_p\left[\mathrm{d}^2 \log p(\mathbf{x}; \mathbf{L})\right] = -\frac{1}{2}\underbrace{\mathrm{d}^2 \log|\mathbf{L}\mathbf{L}^\top|}_{(c)} - \frac{1}{2}\mathbb{E}_p\left[\underbrace{\mathrm{d}^2(\mathbf{x}^\top\mathbf{L}^{-\top}\mathbf{L}^{-1}\mathbf{x})}_{(d)}\right] \tag{25}$$

because $\log|\mathbf{L}\mathbf{L}^\top|$ does not depend on $\mathbf{x}$. By taking advantage of Eq. (21), term $(c)$ is given by

$$
\begin{aligned}
\mathrm{d}^2 \log|\mathbf{L}\mathbf{L}^\top| &= 2\,\mathrm{d}\,\mathrm{tr}(\mathbf{L}^{-1}\,\mathrm{d}\,\mathbf{L}) \\
&= -2\,\mathrm{tr}(\mathbf{L}^{-1}\,\mathrm{d}\,\mathbf{L}\mathbf{L}^{-1}\,\mathrm{d}\,\mathbf{L}) \\
&= -2\,\mathrm{vec}((\mathbf{L}^{-1}\,\mathrm{d}\,\mathbf{L})^\top)^\top\,\mathrm{vec}(\mathbf{L}^{-1}\,\mathrm{d}\,\mathbf{L}) \\
&= -2\left[\mathbf{C}\,\mathrm{vec}(\mathbf{L}^{-1}\,\mathrm{d}\,\mathbf{L}))\right]^\top\,\mathrm{vec}(\mathbf{L}^{-1}\,\mathrm{d}\,\mathbf{L}) \\
&= -2\left[\mathbf{C}(\mathbf{I}\otimes\mathbf{L}^{-1})\,\mathrm{vec}(\mathbf{dL})\right]^\top(\mathbf{I}\otimes\mathbf{L}^{-1})\,\mathrm{vec}(\mathrm{d}\,\mathbf{L}) \\
&= -2\,\mathrm{vec}(\mathbf{dL})^\top(\mathbf{I}\otimes\mathbf{L}^{-\top})\mathbf{C}(\mathbf{I}\otimes\mathbf{L}^{-1})\,\mathrm{vec}(\mathrm{d}\,\mathbf{L}) \\
&= -2\,\mathrm{vec}(\mathrm{d}\,\mathbf{L})^\top\mathbf{C}(\mathbf{L}^{-\top}\otimes\mathbf{L}^{-1})\,\mathrm{vec}(\mathrm{d}\,\mathbf{L}),
\end{aligned}
$$

(26)

(27)

where Eq. (26) follows from $\mathrm{tr}(\mathbf{X}^\top\mathbf{Y}) = \mathrm{vec}(\mathbf{X})^\top\,\mathrm{vec}(\mathbf{Y})$, and $\mathbf{C} = \mathbf{C}^\top$ is the commutator matrix such that $\mathbf{C}\,\mathrm{vec}(\mathbf{X}) = \mathrm{vec}(\mathbf{X}^\top)$ and $\mathbf{C}(\mathbf{X}\otimes\mathbf{Y}) = (\mathbf{Y}\otimes\mathbf{X})\mathbf{C}$.

Similarly, term $(d)$ can be computed starting from the first differential

$$
\begin{aligned}
\mathrm{d}(\mathbf{x}^\top\mathbf{L}^{-\top}\mathbf{L}^{-1}\mathbf{x}) &= \mathbf{x}^\top\,\mathrm{d}(\mathbf{L}^{-\top}\mathbf{L}^{-1})\mathbf{x} \\
&= -\mathbf{x}^\top(\mathbf{L}^{-\top}\,\mathrm{d}\,\mathbf{L}^\top\mathbf{L}^{-\top}\mathbf{L}^{-1} + \mathbf{L}^{-\top}\mathbf{L}^{-1}\,\mathrm{d}\,\mathbf{L}\mathbf{L}^{-1})\mathbf{x} \\
&= -\mathrm{tr}(\mathbf{L}^{-\top}\mathbf{L}^{-1}\mathbf{x}\mathbf{x}^\top\mathbf{L}^{-\top}\,\mathrm{d}\,\mathbf{L}^\top) - \mathrm{tr}(\mathbf{L}^{-1}\mathbf{x}\mathbf{x}^\top\mathbf{L}^{-\top}\mathbf{L}^{-1}\,\mathrm{d}\,\mathbf{L}) \\
&= -2\,\mathrm{tr}(\mathbf{L}^{-1}\mathbf{x}\mathbf{x}^\top\mathbf{L}^{-\top}\mathbf{L}^{-1}\,\mathrm{d}\,\mathbf{L}).
\end{aligned}
$$

(28)

Then, the second differential takes the form

$$
\begin{aligned}
\mathrm{d}^2(\mathbf{x}^\top\mathbf{L}^{-\top}\mathbf{L}^{-1}\mathbf{x}) &= -2\,\mathrm{tr}(\mathrm{d}(\mathbf{L}^{-1}\mathbf{x}\mathbf{x}^\top\mathbf{L}^{-\top}\mathbf{L}^{-1}\,\mathrm{d}\,\mathbf{L})) \\
&= 2\left[\mathrm{tr}(\mathbf{L}^{-1}\,\mathrm{d}\,\mathbf{L}\mathbf{L}^{-1}\mathbf{x}\mathbf{x}^\top\mathbf{L}^{-\top}\mathbf{L}^{-1}\,\mathrm{d}\,\mathbf{L})\right. \\
&\qquad + \mathrm{tr}(\mathbf{L}^{-1}\mathbf{x}\mathbf{x}^\top\mathbf{L}^{-\top}\,\mathrm{d}\,\mathbf{L}^\top\mathbf{L}^{-\top}\mathbf{L}^{-1}\,\mathrm{d}\,\mathbf{L}) \\
&\qquad \left. + \mathrm{tr}(\mathbf{L}^{-1}\mathbf{x}\mathbf{x}^\top\mathbf{L}^{-\top}\mathbf{L}^{-1}\,\mathrm{d}\,\mathbf{L}\mathbf{L}^{-1}\,\mathrm{d}\,\mathbf{L})\right].
\end{aligned}
$$

(29)

By taking the expectation wrt $p$, we have that

$$
\begin{aligned}
\mathbb{E}_p\left[\mathrm{d}^2(\mathbf{x}^\top\mathbf{L}^{-\top}\mathbf{L}^{-1}\mathbf{x})\right] &= 2\left[\mathrm{tr}(\mathbf{L}^{-1}\,\mathrm{d}\,\mathbf{L}\mathbf{L}^{-1}\mathbf{L}\mathbf{L}^\top\mathbf{L}^{-\top}\mathbf{L}^{-1}\,\mathrm{d}\,\mathbf{L})\right. \\
&\qquad + \mathrm{tr}(\mathbf{L}^{-1}\mathbf{L}\mathbf{L}^\top\mathbf{L}^{-\top}\,\mathrm{d}\,\mathbf{L}^\top\mathbf{L}^{-\top}\mathbf{L}^{-1}\,\mathrm{d}\,\mathbf{L}) \\
&\qquad \left. + \mathrm{tr}(\mathbf{L}^{-1}\mathbf{L}\mathbf{L}^\top\mathbf{L}^{-\top}\mathbf{L}^{-1}\,\mathrm{d}\,\mathbf{L}\mathbf{L}^{-1}\,\mathrm{d}\,\mathbf{L})\right] \\
&= 2\left[2\,\mathrm{tr}(\mathbf{L}^{-1}\,\mathrm{d}\,\mathbf{L}\mathbf{L}^{-1}\,\mathrm{d}\,\mathbf{L}) + \mathrm{tr}(\mathrm{d}\,\mathbf{L}^\top\mathbf{L}^{-\top}\mathbf{L}^{-1}\,\mathrm{d}\,\mathbf{L})\right] \\
&= 2\left[2\,\mathrm{vec}(\mathrm{d}\,\mathbf{L})^\top\mathbf{C}(\mathbf{L}^{-\top}\otimes\mathbf{L}^{-1})\,\mathrm{vec}(\mathrm{d}\,\mathbf{L})\right. \\
&\qquad \left. + \mathrm{vec}(\mathrm{d}\,\mathbf{L})^\top\,\mathrm{vec}(\mathbf{L}^{-\top}\mathbf{L}^{-1}\,\mathrm{d}\,\mathbf{L})\right] \\
&= 2\,\mathrm{vec}(\mathrm{d}\,\mathbf{L})^\top\left[2\mathbf{C}(\mathbf{L}^{-\top}\otimes\mathbf{L}^{-1}) + (\mathbf{I}\otimes\mathbf{L}^{-\top}\mathbf{L}^{-1})\right]\mathrm{vec}(\mathrm{d}\,\mathbf{L}),
\end{aligned}
$$

(30)

(31)

where the first component of Eq. (30) follows from the expression already derived in Eq. (27). Overall, the expectation in Eq. (25) becomes

$$
\mathbb{E}_p\left[\mathrm{d}^2\log p(\mathbf{x};\mathbf{L})\right] = -\,\mathrm{vec}(\mathrm{d}\,\mathbf{L})^\top\left[\mathbf{C}(\mathbf{L}^{-\top}\otimes\mathbf{L}^{-1}) + (\mathbf{I}\otimes\mathbf{L}^{-\top}\mathbf{L}^{-1})\right]\mathrm{vec}(\mathrm{d}\,\mathbf{L}),
$$

(32)

which implies that the Fisher information matrix $\mathbf{F}$ takes the form

$$
\begin{aligned}
\mathbf{F} &= -\mathbb{E}_p\left[\frac{\partial^2}{\partial\,\mathrm{vec}\,\mathbf{L}\partial\,\mathrm{vec}\,\mathbf{L}^\top}\log p(\mathbf{x};\mathbf{L})\right] \\
&= \mathbf{C}(\mathbf{L}^{-\top}\otimes\mathbf{L}^{-1}) + (\mathbf{I}\otimes\mathbf{L}^{-\top}\mathbf{L}^{-1}) \\
&= (\mathbf{I}\otimes\mathbf{L}^{-\top})(\mathbf{I}+\mathbf{C})(\mathbf{I}\otimes\mathbf{L}^{-1}).
\end{aligned}
$$

(33)

By rearranging the terms of $Eq.$ (16), we have that the natural gradient may be found without explicitly computing the inverse $\mathbf{F}$ by solving the equation

$$
\mathbf{F}\tilde{\mathbf{g}} = \mathbf{g}
$$

(34)

for $\tilde{\mathbf{g}}$, under the constraint that the latter is the vectorisation of a lower triangular. Before we proceed, let us introduce the matrix form of the the natural gradient vector as $\tilde{\mathbf{G}} = \text{unvec}(\tilde{\mathbf{g}})$. Then we compute

$$
\begin{aligned}
\mathbf{F}\tilde{\mathbf{g}} &= (\mathbf{I} \otimes \mathbf{L}^{-\top})(\mathbf{I} + \mathbf{C})(\mathbf{I} \otimes \mathbf{L}^{-1})\tilde{\mathbf{g}} \\
&= (\mathbf{I} \otimes \mathbf{L}^{-\top})(\mathbf{I} + \mathbf{C})\,\text{vec}(\mathbf{L}^{-1}\tilde{\mathbf{G}}) \\
&= (\mathbf{I} \otimes \mathbf{L}^{-\top})\,\text{vec}(\mathbf{L}^{-1}\tilde{\mathbf{G}} + (\mathbf{L}^{-1}\tilde{\mathbf{G}})^{\top}) \\
&= \text{vec}(\mathbf{L}^{-\top}\mathbf{L}^{-1}\tilde{\mathbf{G}} + \mathbf{L}^{-\top}(\mathbf{L}^{-1}\tilde{\mathbf{G}})^{\top}).
\end{aligned}
\tag{35}
$$

Then, Eq. (34) is equivalent to

$$
\mathbf{L}^{-\top}\mathbf{L}^{-1}\tilde{\mathbf{G}} + \mathbf{L}^{-\top}(\mathbf{L}^{-1}\tilde{\mathbf{G}})^{\top} = \mathbf{A}\mathbf{L} - \mathbf{L}^{-\top}.
\tag{36}
$$

By left-multiplying both sides by $\mathbf{L}^{\top}$, we obtain

$$
\begin{aligned}
\mathbf{L}^{-1}\tilde{\mathbf{G}} + (\mathbf{L}^{-1}\tilde{\mathbf{G}})^{\top} &= \mathbf{L}^{\top}\mathbf{A}\mathbf{L} - \mathbf{I} \\
&= \text{tril}(\mathbf{L}^{\top}\mathbf{A}\mathbf{L}) + \text{triu}(\mathbf{L}^{\top}\mathbf{A}\mathbf{L}) - \text{diag}(\mathbf{L}^{\top}\mathbf{A}\mathbf{L}) - \mathbf{I} \\
&= \text{tril}(\mathbf{L}^{\top}\mathbf{A}\mathbf{L}) - \frac{1}{2}(\mathbf{I} + \text{diag}(\mathbf{L}^{\top}\mathbf{A}\mathbf{L})) \\
&\quad + \left[\text{tril}(\mathbf{L}^{\top}\mathbf{A}\mathbf{L}) - \frac{1}{2}(\mathbf{I} + \text{diag}(\mathbf{L}^{\top}\mathbf{A}\mathbf{L}))\right]^{\top},
\end{aligned}
\tag{37}
$$

which implies that

$$
\mathbf{L}^{-1}\tilde{\mathbf{G}} = \text{tril}(\mathbf{L}^{\top}\mathbf{A}\mathbf{L}) - \frac{1}{2}(\mathbf{I} + \text{diag}(\mathbf{L}^{\top}\mathbf{A}\mathbf{L})).
\tag{38}
$$

Finally, by left-multiplying both sides by $\mathbf{L}$, we find that

$$
\tilde{\mathbf{G}} = \mathbf{L}\left[\text{tril}(\mathbf{L}^{\top}\mathbf{A}\mathbf{L}) - \frac{1}{2}(\mathbf{I} + \text{diag}(\mathbf{L}^{\top}\mathbf{A}\mathbf{L}))\right].
\tag{39}
$$

$\square$

Notice that a similar procedure as the one above can be used to derive the natural gradient of the loss $\ell_{\mathbf{A}}$ with respect to different parameterisations of the covariance matrix of $p(\mathbf{x})$. This includes the marginal parameterisation $p(\mathbf{x}; \mathbf{T}) = \mathcal{N}(\mathbf{0}, \mathbf{T})$, as well as a general square root parameterisation $p(\mathbf{x}; \mathbf{B}) = \mathcal{N}(\mathbf{0}, \mathbf{B}\mathbf{B})$. In particular, it can be shown that, when using the marginal parameterisation, natural gradient descent with unit step size recovers the well-known Newtonian iteration for computing the inverse of a matrix [5].

## B  Proof of Proposition 2

*Proof.* As in the statement of Proposition 2, we use the shorthand $\tilde{\mathbf{K}}_{-} = \mathbf{K}_{\mathbf{uu}} + \tilde{\mathbf{S}}'$, so that $\tilde{\mathbf{K}} = \tilde{\mathbf{K}}_{-} + \sigma^2\mathbf{I}$. Then, proving the inequality

$$
\begin{aligned}
\sigma_n^{2(\mathrm{L})} &= k_{nn} - \mathbf{k}_{n\mathbf{u}}\tilde{\mathbf{K}}^{-1}\mathbf{k}_{\mathbf{u}n} \\
&\geq k_{nn} - \frac{1}{\sigma^2}\left(\mathbf{k}_{n\mathbf{u}}\mathbf{k}_{\mathbf{u}n} - 2\mathbf{k}_{n\mathbf{u}}\mathbf{T}\tilde{\mathbf{K}}_{-}\mathbf{k}_{\mathbf{u}n} + \mathbf{k}_{n\mathbf{u}}\mathbf{T}\tilde{\mathbf{K}}_{-}\tilde{\mathbf{K}}\mathbf{T}\mathbf{k}_{\mathbf{u}n}\right) \\
&= L_n
\end{aligned}
\tag{40}
$$

is equivalent to showing that

$$
\frac{1}{\sigma^2}\left(\mathbf{k}_{n\mathbf{u}}\mathbf{k}_{\mathbf{u}n} - 2\mathbf{k}_{n\mathbf{u}}\mathbf{T}\tilde{\mathbf{K}}_{-}\mathbf{k}_{\mathbf{u}n} + \mathbf{k}_{n\mathbf{u}}\mathbf{T}\tilde{\mathbf{K}}_{-}\tilde{\mathbf{K}}\mathbf{T}\mathbf{k}_{\mathbf{u}n}\right) \geq \mathbf{k}_{n\mathbf{u}}\tilde{\mathbf{K}}^{-1}\mathbf{k}_{\mathbf{u}n}.
\tag{41}
$$

We write $\mathbf{T}\mathbf{k}_{\mathbf{u}n} = \tilde{\mathbf{K}}^{-1}\mathbf{k}_{\mathbf{u}n} + \boldsymbol{\delta}$, where $\boldsymbol{\delta}$ is the discrepancy due to the error with which $\mathbf{T}$ approximates $\tilde{\mathbf{K}}^{-1}$. Notice that both $\tilde{\mathbf{K}}_-$ and $\tilde{\mathbf{K}}$ are PSD. Then, we have

$$\frac{1}{\sigma^2}\left(\mathbf{k}_{n\mathbf{u}}\mathbf{k}_{\mathbf{u}n} - 2\mathbf{k}_{n\mathbf{u}}\mathbf{T}\tilde{\mathbf{K}}_-\mathbf{k}_{\mathbf{u}n} + \mathbf{k}_{n\mathbf{u}}\mathbf{T}\tilde{\mathbf{K}}_-\tilde{\mathbf{K}}\mathbf{T}\mathbf{k}_{\mathbf{u}n}\right)$$

$$= \frac{1}{\sigma^2}\left(\mathbf{k}_{n\mathbf{u}}\mathbf{k}_{\mathbf{u}n} - 2(\tilde{\mathbf{K}}^{-1}\mathbf{k}_{\mathbf{u}n} + \boldsymbol{\delta})^\top\tilde{\mathbf{K}}_-\mathbf{k}_{\mathbf{u}n} + (\tilde{\mathbf{K}}^{-1}\mathbf{k}_{\mathbf{u}n} + \boldsymbol{\delta})^\top\tilde{\mathbf{K}}_-\tilde{\mathbf{K}}(\tilde{\mathbf{K}}^{-1}\mathbf{k}_{\mathbf{u}n} + \boldsymbol{\delta})\right)$$

$$= \frac{1}{\sigma^2}\left(\mathbf{k}_{n\mathbf{u}}\mathbf{k}_{\mathbf{u}n} - \cancel{2}\mathbf{k}_{n\mathbf{u}}\tilde{\mathbf{K}}^{-1}\tilde{\mathbf{K}}_-\mathbf{k}_{\mathbf{u}n} \cancel{-2\boldsymbol{\delta}^\top\tilde{\mathbf{K}}_-\mathbf{k}_{\mathbf{u}n}}\right.$$
$$\left. + \cancel{\mathbf{k}_{n\mathbf{u}}\tilde{\mathbf{K}}^{-1}\tilde{\mathbf{K}}_-\mathbf{k}_{\mathbf{u}n}} \cancel{+2\boldsymbol{\delta}^\top\tilde{\mathbf{K}}_-\mathbf{k}_{\mathbf{u}n}} + \boldsymbol{\delta}^\top\tilde{\mathbf{K}}_-\tilde{\mathbf{K}}\boldsymbol{\delta}\right)$$

$$= \frac{1}{\sigma^2}\left(\mathbf{k}_{n\mathbf{u}}\mathbf{k}_{\mathbf{u}n} - \mathbf{k}_{n\mathbf{u}}\tilde{\mathbf{K}}^{-1}(\tilde{\mathbf{K}} - \sigma^2\mathbf{I})\mathbf{k}_{\mathbf{u}n} + \boldsymbol{\delta}^\top\tilde{\mathbf{K}}_-\tilde{\mathbf{K}}\boldsymbol{\delta}\right)$$

$$= \mathbf{k}_{n\mathbf{u}}\tilde{\mathbf{K}}^{-1}\mathbf{k}_{\mathbf{u}n} + \frac{1}{\sigma^2}\boldsymbol{\delta}^\top\tilde{\mathbf{K}}_-\tilde{\mathbf{K}}\boldsymbol{\delta}$$

$$\geq \mathbf{k}_{n\mathbf{u}}\tilde{\mathbf{K}}^{-1}\mathbf{k}_{\mathbf{u}n}, \tag{42}$$

where the last inequality follow from the fact that the product of two PSD matrices $\tilde{\mathbf{K}}_-$ and $\tilde{\mathbf{K}}$ is also PSD, and with equality when $\boldsymbol{\delta} = \mathbf{0}$, which is implied by $\mathbf{T} = \tilde{\mathbf{K}}^{-1}$. $\qquad\square$

As already discussed in Section 3, the upper bound $U_n$ in Eq. (6) and the lower bound $L_n$ in Eq. (13) that we proved above can be monitored to choose the number of NG optimisation steps for $\mathbf{T}$. In particular, by subtracting $U_n$ and $L_n$, we find the quantity

$$G_n = U_n - L_n$$

$$= k_{n\mathbf{u}}\mathbf{T}\tilde{\mathbf{K}}\mathbf{T}\mathbf{k}_{\mathbf{u}n} - 2\mathbf{k}_{n\mathbf{u}}\mathbf{T}\mathbf{k}_{\mathbf{u}n} + \frac{1}{\sigma^2}\left(\mathbf{k}_{n\mathbf{u}}\mathbf{k}_{\mathbf{u}n} - 2\mathbf{k}_{n\mathbf{u}}\mathbf{T}\tilde{\mathbf{K}}_-\mathbf{k}_{\mathbf{u}n} + \mathbf{k}_{n\mathbf{u}}\mathbf{T}\tilde{\mathbf{K}}_-\tilde{\mathbf{K}}\mathbf{T}\mathbf{k}_{\mathbf{u}n}\right)$$

$$= \frac{1}{\sigma^2}\left(\mathbf{k}_{n\mathbf{u}}\mathbf{k}_{\mathbf{u}n} - 2\mathbf{k}_{n\mathbf{u}}\mathbf{T}(\tilde{\mathbf{K}}_- + \sigma^2\mathbf{I})\mathbf{k}_{\mathbf{u}n} + \mathbf{k}_{n\mathbf{u}}\mathbf{T}(\tilde{\mathbf{K}}_- + \sigma^2\mathbf{I})\tilde{\mathbf{K}}\mathbf{T}\mathbf{k}_{\mathbf{u}n}\right)$$

$$= \frac{1}{\sigma^2}||\mathbf{k}_{n\mathbf{u}} - \tilde{\mathbf{K}}\mathbf{T}\mathbf{k}_{\mathbf{u}n}||^2$$

$$= \frac{1}{\sigma^2}||(\mathbf{I} - \tilde{\mathbf{K}}\mathbf{T})\mathbf{k}_{\mathbf{u}n}||^2. \tag{43}$$

By optimising $\mathbf{T}$ until $G_n < \epsilon'$, one can ensure that the R-SVGP predictive variance in Eq. (6) is at most $\epsilon'$ larger than the L-SVGP predictive variance $\sigma_n^{2(\text{L})}$ in Eq. (4). Moreover, the only part of the SVGP ELBO in Eq. (1) that depends directly on the predictive variance $\sigma_n$ of the parameterisation is the expectation component, which, in the case of Gaussian likelihoods, is given by

$$\sum_{n=1}^{N}\mathbb{E}_{\mathcal{N}(\mu_n,\sigma_n^2)}\left[\log p(y_n|f(\mathbf{x}_n))\right] = -\frac{1}{2\sigma_{\text{obs}}^2}\sum_{n=1}^{N}\sigma_n^2 + c_3, \tag{44}$$

where $\sigma_{\text{obs}}^2$ is the likelihood variance and $c_3$ is a constant that does not depend on $\sigma_n^2$. Therefore, as mentioned in the main text, optimising $\mathbf{T}$ until the stopping criterion

$$\sum_{n=1}^{N}G_n \leq 2\sigma_{\text{obs}}^2\epsilon \tag{45}$$

is reached ensures that the slack in the ELBO of the R-SVGP parameterisation introduced by upper bounding the predictive variance $\sigma_n^{2(\text{L})}$ of the L-SVGP parameterisation is at most $\epsilon$.

The careful reader may have noticed that the stopping criterion derived above resembles a commonly used approach for iterative GP approximations [11, 7]. In particular, iterative frameworks for solving GPs approximate key quantities of exact GPs, such as their predictive mean

$$\mu_* = \mathbf{k}_{*\mathbf{u}}\mathbf{K}_{\mathbf{ff}}^{-1}\mathbf{y}, \tag{46}$$

by using iterative solvers (e.g. conjugate gradients) for the linear system $\mathbf{K}_{\mathbf{ff}}\mathbf{x} = \mathbf{y}$, until a stopping criterion is reached. A common criterion [11] monitors the slack due to the approximation in the quadratic term of the true GP log marginal likelihood, i.e.

$$-\frac{1}{2}\mathbf{y}^\top\mathbf{K}_{\mathbf{ff}}^{-1}\mathbf{y}, \tag{47}$$

by employing upper and lower bounds on Eq. (47), which resemble the bounds $U_n$ and $L_n$, with $\mathbf{k}_{\mathbf{u}n}$ replaced by $\mathbf{y}$.

## C   Experimental details

### C.1   Datasets

For the experiments presented in Section 4 and Appendix D, we use datasets from three sources:

- **UCI repository**[1]: the ELEVATORS ($N = 16599$, $D = 18$, $B = 1000$), KEGGDIRECTED ($N = 48827$, $D = 20$, $B = 1000$), KEGGUNDIRECTED ($N = 63608$, $D = 27$, $B = 1000$) and KIN40K ($N = 40000$, $D = 8$, $B = 1000$) regression datasets;
- **OpenML**[2].: the BANANA ($N = 5300$, $D = 2$, $B = 64$) classification dataset;
- **Other**: the SNELSON ($N = 200$, $D = 1$, $B = 10$) regression dataset [19].

For the SNELSON and BANANA datasets we only consider the training fit on the whole dataset, whereas for the UCI datasets we perform a 90/10 train/test split.

### C.2   Model setup

For all the bounds and throughout all the experiments:

- We use the ARD squared exponential kernel initialised using the GPFlow defaults;
- We initialise the inducing locations $\mathbf{Z}$ drawing uniformly without replacement from the training observations;
- We initialise the Gaussian (regression) or Bernoulli (classification) likelihood using the GPFlow defaults.

By default, GPFlow applies whitening to the M-SVGP parameterisation. In particular, the inducing mean and variance are reparameterised, respectively, as $\mathbf{m} = \mathbf{L}_{\mathbf{uu}}\tilde{\mathbf{m}}$ and $\mathbf{S} = \mathbf{L}_{\mathbf{uu}}\tilde{\mathbf{S}}\mathbf{L}_{\mathbf{uu}}^{\top}$, where $\mathbf{L}_{\mathbf{uu}}$ is the Cholesky factor of $\mathbf{K}_{\mathbf{uu}}$. In experiments not reported here, we found M-SVGP with whitening to generally exhibit better optimisation performance than standard M-SVGP on the datasets used in this work. For this reason, the results for M-SVGP in Section 4 and in Appendix D refer to the whitened version.

Furthermore, depending on the bound, we initialise the inducing parameters as follows:

- M-SVGP: $\tilde{\mathbf{m}}$ and $\tilde{\mathbf{S}}$ are initialised using the GPFlow defaults;
- L-SVGP: we initialise $\tilde{\mathbf{m}} = \mathbf{0}$ and $\tilde{\mathbf{S}} = \sigma_{\tilde{\mathbf{S}}}^2 \mathbf{I}$ with $\sigma_{\tilde{\mathbf{S}}}^2 = 1^{-4}$;
- R-SVGP: we initialise $\tilde{\mathbf{m}}$ and $\tilde{\mathbf{S}}$ as with L-SVGP, and $\mathbf{T} = \sigma_{\mathbf{T}}^2 \mathbf{I}$ with $\sigma_{\mathbf{T}}^2 = 1^{-6}$.

### C.3   Training

The mini-batch sizes $B$ are listed in Appendix C.1, while the number of training iterations is either clear from the figures or mentioned in each specific experimental section in Appendix D, when needed. The learning rate of Adam is kept fixed during training to $1^{-2}$ for the BANANA dataset, and to $5^{-3}$ everywhere else. Lastly, the schedule of the NG updates for $\mathbf{T}$ is as follows:

- UCI datasets with $\mathbf{Z}$ trainable: the initial step size of $1^{-2}$ is doubled after every NG update until the stopping criterion $G$ falls below the threshold values of $1.0$ for ELEVATORS and KIN40K, and $1^{-3}$ for KEGGDIRECTED and KEGGUNDIRECTED;
- Everywhere else: a single NG update with step size $1.0$ is used.

---

[1]https://archive.ics.uci.edu/datasets
[2]https://www.openml.org/search?type=data

## C.4 Implementation

We implement our methods using GPFlow [13], which is based on Tensorflow [1]. All the experiments were run on a single NVIDIA V100 GPU with 16GB of memory.

# D    Additional results

## D.1    Quality of learned $\mathbf{T}$

We aim to evaluate the effectiveness of Adam and NG in keeping the matrix $\mathbf{T}$ close to its optimal value throughout training. To do so, we consider the KIN40K dataset with $M = 1000$ and initialise $\mathbf{T}$ to its optimal value $\tilde{\mathbf{K}}^{-1}$. Then, we train the R-SVGP and R-SVGP-N bounds for $100000$ iterations. Fig. 2 shows the quality of the $\mathbf{T}$ matrix learned by the two methods during training, as measured by $\mathrm{KL}[\mathcal{N}(\mathbf{0}, \mathbf{LL}^{\top}), \mathcal{N}(\mathbf{0}, \mathbf{A}^{-1})]$, which is the NG objective for $\mathbf{T}$. As it can be clearly seen, the NG

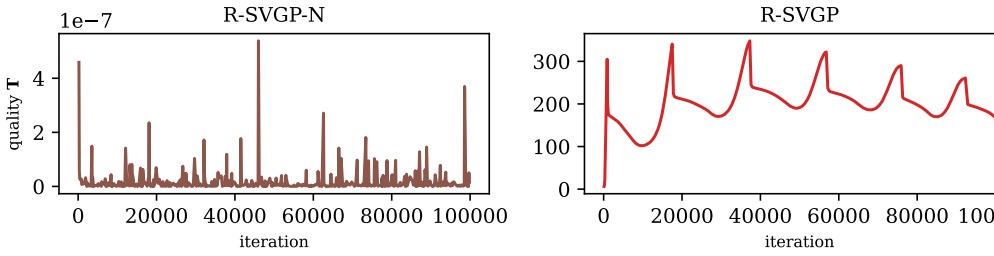

Figure 2: Quality of the $\mathbf{T}$ matrix learned during training by optimising the RSVGP bound with NG (left) and with Adam (right).

updates are able to consistently keep $\mathbf{T}$ close to its optimal value, whereas this is not the case for Adam. In particular, the R-SVGP bound exhibits the same erratic behaviour observed by van der Wilk et al. [22], where the quality of $\mathbf{T}$ repeatedly improves until a point where it suddenly gets worse, hampering effective training.

## D.2    UCI datasets with fixed inducing locations

We evaluate the ability of our method to optimise the R-SVGP bound beyond toy datasets by testing it on four larger UCI regression datasets. In particular, we use each of the bounds discussed in Section 4 to train a sparse variational GP model with fixed inducing locations $\mathbf{Z}$ for 20000 iterations following the specifics described in Appendix C.

Fig. 3 shows the training loss and predictive metrics (RMSE and NLPD) achieved by the models at the end of training for four choices of the number of inducing points $M$ ranging from 1000 and 4000. As it can be seen, each of the R-SVGP bounds trained with NG (RSVGP-N and RSVGP-NP) consistently matches the performance of the corresponding L-SVGP bound (LSVGP and LSVGP-P, respectively). In particular, the R-SVGP-NP bound achieves competitive performance compared to the standard M-SVGP parameterisation.

As reported by van der Wilk et al. [22], we find that the R-SVGP bounds trained only with Adam (i.e. R-SVGP and R-SVGP-P) exhibit erratic performance, and we report their results separately in Fig. 4.

## D.3    UCI datasets with trained inducing locations

We repeat the experiments in Appendix D.2, while also training the inducing locations $\mathbf{Z}$ as part of the Adam step. In this case, the optimisation of $\mathbf{T}$ in the inverse-free bounds is more challenging because, when $\mathbf{Z}$ moves, we expect the optimal value of $\mathbf{T}$, $\tilde{\mathbf{K}}^{-1} = (\mathbf{K_{uu}} + \tilde{\mathbf{S}})^{-1}$, to change more between iterations compared to when $\mathbf{Z}$ is fixed. For this reason, for R-SVGP-N and R-SVGP-NP, we perform multiple NG updates at each iteration according to the schedule described in Appendix C.3.

Fig. 5 shows the training loss and predictive metrics (RMSE and NLPD) achieved by the models at the end of training for four choices of the number of inducing points $M$ ranging from 1000 and

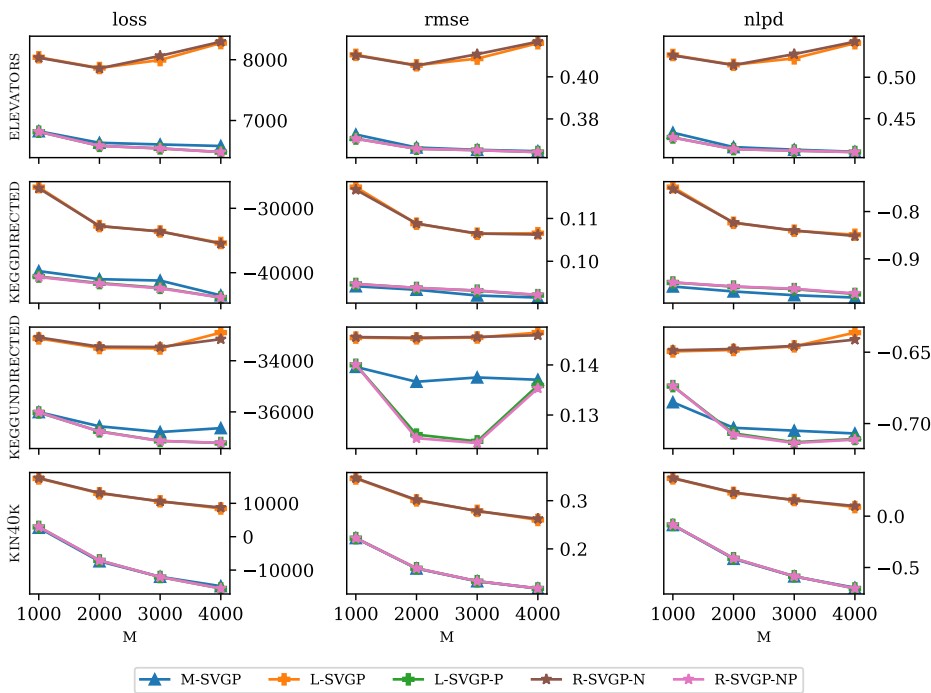

Figure 3: Final loss (negative ELBO) and predictive metrics (RMSE and NLPD) on four UCI datasets for different choices of $M$ with fixed $\mathbf{Z}$. The legend keys are as in Fig. 1.

4000. In this case, training $\mathbf{Z}$ introduces a bit more variance between runs compared to the case where $\mathbf{Z}$ is fixed. We believe that using a more refined scheduling for the step-size of the NG updates, such as commonly used log-linear scheduling [18], may help to reduce it. In any case, the main takeaway remains the same: the R-SVGP-NP bound achieves competitive performance compared to the standard M-SVGP parameterisation even when $\mathbf{Z}$ is trained.

As before, the results for the inverse-free bounds trained only with Adam (R-SVGP and R-SVGP-P) are unstable, and we report them separately in Fig. 6.

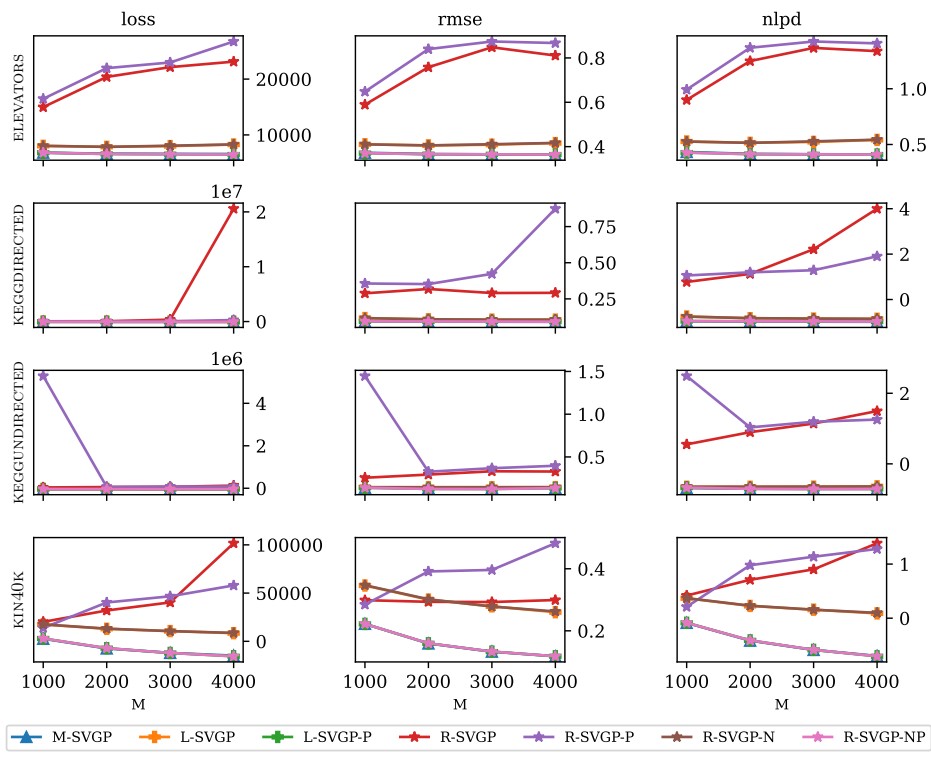

Figure 4: Same content as in Fig. 3, with the addition of the RSVGP bounds trained solely with Adam, both with (R-SVGP-P) and without (R-SVGP) inducing mean preconditioning.

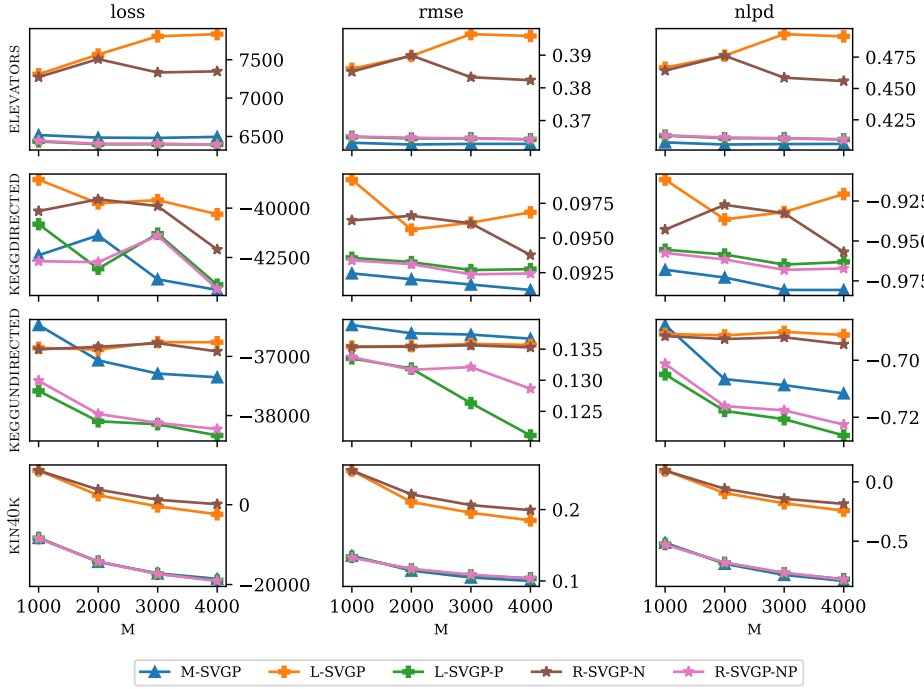

Figure 5: Final loss (negative ELBO) and predictive metrics (RMSE and NLPD) on four UCI datasets for different choices of $M$ with trainable $\mathbf{Z}$. The legend keys are as in Fig. 1.

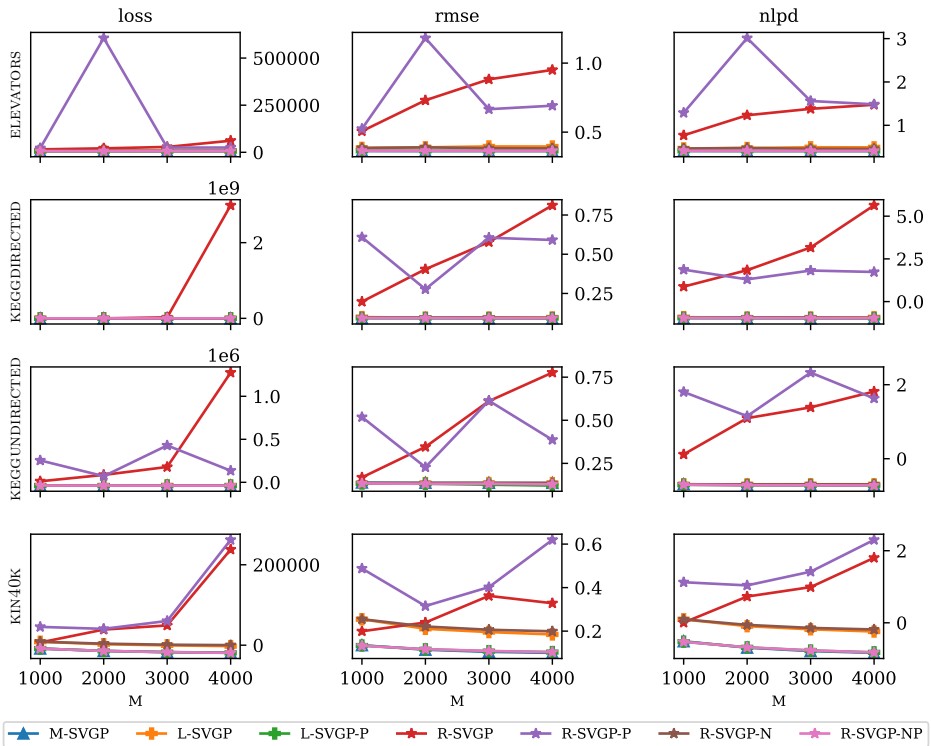

Figure 6: Same content as in Fig. 5, with the addition of the RSVGP bounds trained solely with Adam, both with (R-SVGP-P) and without (R-SVGP) inducing mean preconditioning.

