# OpenReview forum: "Inverse-Free Sparse Variational Gaussian Processes"
_NeurIPS.cc/2024/Workshop/BDU — NeurIPS BDU Workshop 2024 Poster_

### Official Review · Reviewer_wunz · 2024-09-26

**Rating:** 7
**Confidence:** 4

**Review:**

This paper presents a novel approach to sparse variational Gaussian processes (SVGP) by introducing an inverse-free optimization method, which is a meaningful contribution. Using matrix-free natural gradient updates to avoid kernel matrix inversions is particularly relevant for large-scale GPs. The following are my comments:

1.	I would suggest adding a computational complexity analysis for the natural gradient update approach, detailing how the proposed method scales with the number of inducing points and the data size in contrast to traditional inversion-based methods.
2.	The authors mentioned that natural gradients converge faster, but is there any potential instability or sensitivity in the choice of step size for the natural gradient? For instance, are there scenarios where convergence could be slower due to poor conditioning of T?
3.	The stopping criterion for the natural gradient updates seems sound, but how did the authors choose the threshold and how sensitive is the method to different thresholds?
4.	I would suggest making more explicit comparisons to other inversion-based methods in terms of both accuracy (e.g., RMSE) and computational performance (e.g., iteration count or time to convergence).
5.	Please discuss the practical limitations of the proposed method in certain use cases (e.g., how it might perform in extremely high-dimensional or non-stationary data).

---

### Official Review · Reviewer_NcAD · 2024-09-28

**Rating:** 7
**Confidence:** 3

**Review:**

The paper proposes an *inverse-free* variational approach for sparse Gaussian Processes (GPs), focusing on optimizing a new loss function that avoids matrix inversion. The method incorporates natural gradients and preconditioning techniques, demonstrating effective performance across various datasets.

**Strengths:**
- Interesting and Well-Motivated Loss Function: The newly proposed loss function is a novel contribution that significantly improves the optimization process for GPs. The combination of natural gradient updates and preconditioning provides a clear advantage over traditional methods.

- Solid Validation: The paper includes thorough empirical validation across toy datasets and real-world datasets, demonstrating that the proposed approach is on par or even superior to existing methods in terms of performance.

---

### Decision · Program_Chairs · 2024-10-09

Accept (Poster)